# In Vitro Antimicrobial Effects and Inactivation Mechanisms of 5,8-Dihydroxy-1,4-Napthoquinone

**DOI:** 10.3390/antibiotics11111537

**Published:** 2022-11-03

**Authors:** Seray Topçu, Mine Gül Şeker

**Affiliations:** Department of Molecular Biology and Genetics, Gebze Technical University, Gebze 41 400, Kocaeli, Türkiye

**Keywords:** antimicrobial effects, MIC_50_, 5,8-dihydroxy-1,4-naphthoquinone, inactivation mechanisms

## Abstract

Naphthoquinones are an important class of natural organic compounds that have antimicrobial effects. However, the mechanisms of their action remain to be elucidated. Therefore, the antimicrobial activity of the chemically synthesized naphthoquinone derivative, 5,8-dihydroxy-1,4-naphthoquinone, was investigated in this study against 10 different microorganisms. Its inhibitory activity was evident against *Bacillus cereus*, *Proteus vulgaris*, *Salmonella enteritidis*, *Staphylococcus epidermidis*, *S. aureus*, and *Candida albicans*, and its MIC_50_ values were determined to be 14, 10, 6, 2, 4, 1.2, and <0.6 µg/mL, respectively. Moreover, the crystal violet uptake, TTC dehydrogenase activity, protein/DNA leakage, and DNA damage of the compound in these microorganisms were also investigated to reveal the antimicrobial mechanisms. In addition, scanning electron microscopy was used to detect physiological damage to the cell membrane of *S. epidermidis*, *S. aureus*, and *C. albicans*, which was most severe in the crystal violet uptake assay. The overall results showed that 5,8-dihydroxy-1,4-naphthoquinone exhibited its effects on *S. aureus*, *S. epidermidis*, and *C. albicans* by various mechanisms, especially membrane damage and membrane integrity disruption. It also caused DNA leakage and damage along with respiratory chain disruption (78%) in *C. albicans*. Similarly, it caused varying degrees of reduction in the respiratory activity of *S. aureus* (47%), *S. epidermidis* (16%), *B. cereus* (12%), *S. enteritidis* (9%), and *P. vulgaris* (8%). Therefore, 5,8-dihydroxy-1,4-naphthoquinone proved to be a very effective antifungal and antibacterial agent and could be considered a new potential drug candidate, inspiring further discoveries in these microorganisms.

## 1. Introduction

Naphthoquinones (NQ) are derived from naphthalene and are known as bicyclic molecules consisting of two rings containing carbonyl groups in two different positions (1, 4 or 1, 2) [1]. The central molecule of naphthalene consists of a benzene moiety fully conjugated to another cyclic diketone ring. Due to these properties, 1,4-NQs are produced by various horticultural plants; in addition, some fungi and animals [2] have antibacterial [3], antimalarial [4] antiparasitic [5], antiviral, and antifungal activities [6,7,8] and thus have long been used in traditional medicine [9,10]. Moreover, the substitution of some side groups to NQs, such as hydroxyl, methyl, nitrogen, sulfur, halide, phenylamino-phenylthio, sulfur, etc., has conferred additional remarkable biological activities to the molecule [10]. In particular, NQs with hydroxyl groups, such as naphthazarin (5,8-dihydroxy-1,4-NQ), which is one of the natural substance that are derived from the tissues of several members of the Boraginaceae, Nepenthaceae, and Droseraceae families [11], have remarkable pharmacological activities compared to other derivatives [12]. Therefore, this NQ derivative with its special structural (Appendix A), biological, and functional properties are considered good candidates for medicinal chemistry [12] and are also commercially produced.

Previously, the antimicrobial and antifungal activities of various natural and synthetic NQs, including 5-methoxy-3,4-dehydroxanthomegnine, 5-hydroxy-3,6-dimethoxy-2-methylnaphthalene-1,4-dione and 5,8-Dihydroxy-3-methoxy-2-methylnaphthalene-1,4-dione, 1,4-naphthoquinone-[3,2-c]-1H-pyrazoles, and their 1,4-naphthohydroquinone derivatives ρ-anisidilo, σ-anisidilo, phenyl and methyl, 2-hydroxy-1, 4- naphthoquinone derivatives with cyano and 4-chlorophenyl moieties in C4, S-, S, N-, and N, S-1,4-NQ derivatives, 2-ethylamino-3-methyl-1,4-NQ are found on various microorganisms such as *Helicobacter pylori* [13], *Mycobacterium tuberculosis* [14], *Streptococcus* (*Enterococcus*) *faecalis*, *Klebsiella pneumoniae*, *Escherichia coli* [6], *Staphylococcus. aureus* [15], *E. coli*, *S. aureus*, *M. luteum* [16], *Pseudomonas aeruginosa*, and *S. aureus* [17]. However, the mechanism of their action against these microorganisms has not been elucidated, as only limited studies have been reported to evaluate the mechanism for the antibacterial effect of NQs, which is attributed to the inhibition of electron transfer in the mitochondrial respiratory chain and thus the production of ROS and radical semiquinones [18,19]. Moreover, the production of ROS is also associated with apoptosis [20]. However, it is well-known that the mechanism of inhibition can be obtained by studying the uptake, damage, and leakage of intracellular components (DNA, proteins, etc); the disruption of cell homeostasis; the effects on the membrane; and the inhibition of enzymes of the electron transport system and oxidative phosphorylation and can be visualized by the microscopic imaging (scanning or transmission electron microscopy, SEM/TEM) of cell structure and various chemical/synthetic compounds or natural plant extracts of compounds [20]. Thus, the aims of this study concerned (i) the determination of the antibacterial activity of commercial 5,8-dihydroxy-1,4-NQ (Apollo Scientific, USA) on standard strains of various Gram-negative (*Escherichia coli* ATCC 8739, *Pseudomonas aeruginosa* ATCC 15692, *Proteus vulgaris* (laboratory isolate) *Klebsiella pneumoniae* (laboratory isolate), *Salmonella enteritidis* (laboratory isolate)) and Gram-positive (*Enterococcus faecalis* ATCC 8213, *Staphylococcus. aureus* ATCC 29213, *Staphylococcus epidermidis* ATCC 12228, *Bacillus cereus* DSMZ 4312) bacteria and a yeast *(Candida albicans* ATCC 10231); (ii) the determination of the minimum inhibition concentration (MIC_50_) for sensitive microorganisms; (iii) the evaluation of the mechanism of action of the compound by cell membrane permeability (using crystal violet assay), DNA double strand damage (using SYBR green I) and leakage (at 260 nm wavelength), protein leakage (at 595 nm wavelength) assays, and visualization by SEM for selected microorganisms.

## 2. Results

### 2.1. Antimicrobial Test

5,8-Dihydroxy-1,4-NQ showed moderate to high microbial activity in all tested microorganisms with zone diameters of 8–30 mm compared to the zone diameters (18–26 mm) of the control antibiotic disk [Chloramphenicol (C^30^)] for the tested microorganisms except *K. pneumoniae and P. aeruginosa* (Table 1). In addition, *E. coli* and *E. faecalis* were not used for further experiments because they did not form a strong zone compared to the zone diameter of the controls (Figure 1, Table 1).

Zone diameters of the other strains ranged from 30 mm to 8 mm. With a zone diameter of 30 mm, *S. epidermidis* had the highest zone diameter compared to the results of the control antibiotic disc (26 mm). In addition, *P. vulgaris* had a higher zone diameter (19 mm) than the C^30^ zone diameter.

### 2.2. Minimum Inhibition Concentration (MIC_50_)

The MIC_50_ values (concentration that inhibits half of the cell cultures) of the *C. albicans*, *S. aureus*, *B. cereus*, *P. vulgaris*, *S. epidermidis*, and *S. enteritidis* bacteria are shown in Table 2. 

MIC_50_ values of 5,8-dihydroxy-1,4-NQ against selected standard microorganisms were determined to be *B. cereus* 14 μg/mL, *P. vulgaris* 10 μg/mL, *S. enteritidis* 6 μg/mL, *S. epidermidis* 2.4 μg/mL, and *S. aureus* 1.2 μg/mL. Among all MIC_50_ values, the value of *C. albicans* was very low (<0.6 μg/mL).

### 2.3. Crystal Violet Assay

While the percentage of crystal violet remaining in the supernatants of empty crystal violet (eCV) was assumed to be approximately 100%, the percentage value of the amount of crystal violet remaining in the supernatants of treated cells was calculated. Then the difference between eCV and remaining CV was noted as the amount of intracellular uptake (%). Specifically, it was shown that the cell membrane integrity of *S. aureus* and *C. albicans* was damaged by nearly 91.08% and 61.39%, respectively (Figure 2). These were followed by *S. epidermidis*, *P. vulgaris*, *B. cereus*, and *S. enteritidis*, with 47.36%, 30.08%, 26.88%, and 13.02%, respectively.

### 2.4. Measurement of TTC Dehydrogenase Activity

The relative TTC dehydrogenase activity (RDA) of the standard microorganisms was calculated against 5,8-dihydroxy-1,4-NQ. The RDA percentages of the untreated cells were taken as 100%. The relative TTC dehydrogenase activity of *C. albicans*, *S. aureus*, *S. epidermidis*, *B. cereus*, *S. enteritidis*, and *P. vulgaris* was decreased by 78.07%, 47.42%, 16.05%, 12%, 9.4%, and 8.11%, respectively (Figure 3).

### 2.5. DNA Leakage

DNA leakage was measured at 260 nm using a spectrophotometer; 260 nm (OD_260_) = 1 was accepted as 50 µg/mL DNA leakage. According to measurement results, only a small amount of DNA was detected in the supernatants of *S. enteritidis* (4.375 µg/mL), *C. albicans*. 3.25 µg/mL), and *S. epidermidis* (2.875 µg/mL). DNA leakage was not detected for other tested microorganisms (*S. aureus*, *B. cereus*, and *P. vulgaris*) (Table 3).

### 2.6. Protein Leakage

Protein content in the supernatants of the samples was measured at 595 nm. The OD values of the samples were exactly the same as those of the control.

### 2.7. SDS-PAGE Analysis

Samples of selected microorganisms treated with 5,8-dihydroxy-1,4-NQ do not lead to any differentiation in protein profile in comparison to negative controls.

### 2.8. DNA Damage

5,8 Dihydroxy-1,4 NQ caused damage to the DNA double-strand of the *C. albicans* samples (Figure 4f), whereas it did not show significant differences in *S. aureus*, *B. cereus*, *P. vulgaris*, *S. epidermidis*, or *S. enteritidis* compared to the negative controls (Figure 4a–e).

### 2.9. FT–IR Analysis

No significant difference was found in the peaks determined by FT–IR analysis between the spectra of cells treated with 5,8-dihydroxy-1,4-NQ and untreated control cells.

### 2.10. SEM Analysis

Physiological damage of the cell membrane of three selected bacteria was demonstrated by SEM. SEM images of treated and control *S. aureus*, *S. epidermidis*, and *C. albicans* are shown in Figure 5, Figure 6, and Figure 7, respectively. As shown in Figure 5a,c,e, untreated *S. aureus* were typically round-shaped in coccus form with smooth and intact cell walls. After treatment with 5,8-dihydroxy-1,4-NQ, the morphology of *S. aureus* was deeply effected, as the degradation of individual cell membranes was visualized, and the cell shapes became wrinkled and damaged (Figure 5b,d,f).

Differently, an opaque texture on the pellet (Figure 6b) was observed in 5,8-dihydroxy-1,4-NQ-treated *S. epidermidis* compared to the untreated control (Figure 6a). Moreover, the smooth (Figure 6e) cell surface became rough after treatment with the compound (Figure 6f).

The comparison of SEM images of 5,8-dihydroxy-1,4-NQ-treated *C. albicans* pellets (Figure 6b) with control cells (Figure 6a) clearly revealed disturbances in pellet structure (Figure 7c,d) with treatment. In addition, the highest magnification (2.00 µm) of SEM images demonstrated the healthy round shape of *C. albicans* controls (Figure 7e), while cracks on cell membrane were observed in 5,8-dihydroxy-1,4-NQ-treated cells (Figure 7e in square).

## 3. Discussion

Since the substitution of various functional groups in NQs resulted in new derivatives with better activity than commercial antibacterial agents [21], this study investigated the antimicrobial activity and mechanisms of action of commercial 5,8-dihydroxy-1,4-NQ on various microorganisms. According to the results of the screening test, the compound was found to be highly effective on some tested Gram-positive and Gram-negative bacteria and especially on *C. albicans* as yeast (Figure 1). Interestingly, however, the results of the antibacterial screening test and the MIC_50_ values of the microorganisms treated with the compound were relatively different, such as *P. vulgaris* (19 mm zone diameter, 10 µg/mL MIC_50_ value) or *C. albicans* (18 mm zone diameter, <0.6 μg/mL MIC_50_ value). These differences could be due to the utilization of MHA and SDA in the agar well method. Although commercially powdered 5,8-dihydroxy 1,4-NQ is soluble in DMSO, it may disperse more poorly on the MHA surface than on the SDA medium surface. Furthermore, it should be noted that the concentration used for the screening test was high (1% *w*/*v*), but the MIC_50_ value corresponds to the amount that kills half of the added microorganisms compared to decreasing concentrations.

In the literature, 1,4-NQ and plumbagin (5-hydroxy-2-methyl-1,4-NQ) at concentrations ˂1 mg/mL were found to be effective against both Gram-positive (*B. cereus* ATCC 14579, *Micrococcus luteus* ATCC 4698, methicillin-susceptible *Staphylococcus aureus* (MSSA) ATCC 25923, MRSA ATCC 33591, MRSA clinical strain 1, MRSA clinical strain 2, *E. faecalis* ATCC 29212) and Gram-negative bacteria (*P. aeruginosa* ATCC 10145, *K. pneumoniae* ATCC 10031, *K. pneumoniae* ATCC 1031, *K. pneumoniae* ATCC 2146, *K. pneumoniae* ATCC 1705, *E. coli* ATCC 25922, *Salmonella choleraesuis* ATCC 10708) [22]. Other study results reported that Lawsone (2-hydroxy-1,4-NQ derivative) had anti-tumor and anti-fungal effects, but the MIC_50_ value of this compound against *S. aureus* ATCC 29213 (methicillin-susceptible *S. aureus*) was relatively high (~32 μg/mL), with no selectivity [23] compared to our results with 5,8-dihydroxy-1,4-NQ for the same strain (1.2 µg/mL). In contrast, Wellington et al. (2019) studied sulfide-substituted 1,4-NQ derivatives and reported MIC_50_ levels for *S. aureus* of 7.8 μg/mL, whereas it was 23.4 μg/mL for *C. albicans*. The MIC_50_ value for *Candida* spp was decreased to 2–4 μg/mL by the incorporation of chlorine subgroups in 5,8-dihydroxy-1,4-NQ [2,7]. These MIC_50_ values even decreased to <0.6 μg/mL with the substitution of two hydroxyl subgroups to 1,4-NQ, showing that the inclusion of hydroxyl side groups in the same compound produced a stronger antifungal effect than the previously synthesized derivatives. This finding was also true for *S. aureus*, *S. epidermidis*, and *B. cereus* as relatively lower MIC_50_ values (1.2 µg/mL, 2.4 µg/mL, 14 µg/mL, respectively) were obtained compared to previous reports with other NQ derivatives such as 2-hydroxy-1,4-NQ derivatives containing cyane and 4-chlorophenyl moieties in C4 (MIC_50_ 16.0–64.0 µg/mL for *S. aureus*, [15]) and nickel-, chromium-, iron-, copper-, and cobalt-containing metal chelates of 5-amino-8-hydroxy-1,4-NQ and its acyl-derivatives (MIC_50_ 50–1050 µg/mL for *S. epidermidis*; 125–1400 µg/mL for *B. cereus* [24]). This increased antimicrobial activity may be due to the enhancement of the hydrogen bonding, which allows stronger binding at the site of action by substituting dihydroxyl groups at the 5 and 8 positions [24].

The mechanism of action of 5,8-dihydroxy-1,4-NQ on membrane permeability and integrity was investigated by crystal violet uptake and SEM analysis. Our results showed that 5,8-dihydroxy-1,4-NQ disrupted the membrane permeability of *S. aureus*, *C. albicans*, *S. epidermidis*, *P. vulgaris*, *B. cereus*, and *S. enteritidis* by 91.83%, 64.61%, 47.36%, 30.08%, 26.88%, and 13.02%, respectively (Figure 2). According to the membrane permeability results (crystal violet uptake (%)), 5,8-dihydroxy-1,4-NQ was effective even at low concentrations for three microorganisms, and the results were consistent with the MIC_50_. Although *C. albicans* is also a eukaryote, it has a thick cell wall and a cytoplasmic membrane like Gram-positive bacteria. It is possible that NQ increases the permeability of this thick cell wall by affecting peptidoglycan (in prokaryotes), chitin, glucan, and/or mannoproteins (in fungi), as found by Futuro and colleagues (2018)’s report with 23 different synthetic mannic base derivatives from Lawsone by increasing the membrane permeability of *C. albicans* ATCC 10231 [25]. Similarly, Shrestha and colleagues (2017) reported the membrane disruption of some yeasts such as *C. albicans* ATCC 64124 and *Fusarium graminearum* B4-A5 by Sytox Green dye with the synthesized cationic amphiphiles resembling quaternary ammonium complexes (QAC) based on 1,4 NQ [26]. Moreover, Do Perpetuo and colleagues (2014) stated that the tested synthetic NQ compounds altered the cell wall of the fungus and showed their effect by damaging the membrane permeability of the cell [27]. In contrast, the 5,8-dihydroxy-1,4-NQ used in this study showed its effect by damaging the membrane at much lower concentrations (<0.6 μg/mL). In collaboration, cell membrane damage was also demonstrated in *C. albicans*, *S. epidermidis*, and *S. aureus* in SEM analysis.

In addition, the antimicrobial effects of NQ are also attributed to the inhibition of electron transport or damage to oxidative phosphorylation in the respiratory system. Many quinone derivatives cause the stimulation of superoxide production in bacterial cells [11], which affects cells survival. In this study, the metabolic activity of cells was measured using TTC dehydrogenase, which is part of the electron transfer system on the bacterial membrane (Figure 3). The activity of the TTC dehydrogenase of selected microorganisms (*C. albicans* (21.93%), *S. aureus* (52.57%), *S. epidermidis* (83.95%), *B. cereus* (88%), *S. enteritidis* (90.57%), *P. vulgaris* (91.89%)) showed different decrease percentages in survival. Moreover, the inhibition of the TTC dehydrogenase (21.93%) of *C. albicans* appeared to be less effective than other mechanisms. It should be taken into account that 5,8-dihydroxy-1,4-NQ showed this effect on *C. albicans* at very low doses (<0.6 μg/mL), and it seemed that the presence of respiratory enzymes in the inner mitochondrial membrane in *C. albicans* could cause these enzymes to be less affected. In contrast, in bacteria, the electron transport chain enzymes are located in the cell membrane, which may lead them to be directly affected by NQ. As a result, the 5,8-dihydrox-1,4-NQ compound affects the hydrogen transport capacities of dehydrogenase enzymes and reverses the oxidation property in all treated cells.

Undoubtedly, DNA and protein leakage are reference points that can be associated with membrane damage. However, in the current study, no protein leakage was detected in the supernatants of the tested microorganisms treated with 5,8-dihydroxy-1,4-NQ compared with controls. On the contrary, in agreement with the damage in the crystal violet uptake assay and the images from SEM, only in *S. epidermidis* and *C. albicans* was an extracellular DNA leakage of 2.875 µg/mL and 3.25 µg/mL, respectively, detected by spectroscopic measurement. The presence of a break on the membrane (Figure 7) in the SEM images of *C. albicans* also explains the leakage of DNA in that microorganism. The other tested microorganisms, wherein the leakage of proteins (in all tested microorganisms) and DNA (*S. aureus*, *B. cereus*, *P. vulgaris*) cannot be measured, may demonstrate that intracellular proteins larger than DNA do not leak through the cell membrane. If the Na^+^/K^+^ pumps are damaged, this suggests that electrolytes (such as K^+^, Cl^−2^, Mg^+2^) may leak out because they are smaller than DNA and proteins. While there are studies in the literature on the leakage of electrolytes, proteins, and DNA during treatment with antimicrobial agents, such as manganese dioxide nanomolecules [28], disinfectants [29], and electro-active water [30], no study has been conducted on the leakage of ions as a result of treatment with NQ. Therefore, it is predicted that if the concentration of 5,8-dihydroxy-1,4-NQ compound is increased, the leakage of DNA and other materials might increase.

When the protein profile of the negative control was compared with the protein profiles of cells treated with the 5,8-dihydroxy-1,4-NQ, no significant difference was found. However, it is known that the SDS-PAGE method sometimes does not have adequate sensitivity to detect very small changes in membrane proteins [31]. Thus, if minor changes in protein structure, such as the breaking of amide bands and C–H bonds of amino acids, occurred as a result of the compound treatment, the SDS-PAGE method might not be sufficient.

SYBR green dye is a fluorescent dye used to detect DNA double-strand damage. It causes an emission that can be measured spectrophotometrically at an absorbance of 525 nm. Breaks and damage that occur in DNA after the application of a chemical agent cause a decrease in the measurable signal. Due to this property, it is used as a marker that can be used in the detection of DNA damage [30]. In the present study, cells stained with 5,8-dihydroxy-1,4-NQ along with control microorganisms were stained with SYBR Green I, and then the absorbance was measured spectroscopically at 525 nm (OD_525_). The results showed that *C. albicans* had the most DNA damage compared its own control (0.505/0.341 absorbance) (Figure 4). In addition, a small amount of DNA damage was also observed in *S. enteritidis*, but it was not statistically significant. Moreover, alterations in protein function, the disruption of mitochondrial activity, and the inhibition of enzyme and membrane damage are known to cause the accumulation of ROS in *Candida* spp. [32]. On the other hand, quinones, which include 5,8-dihydroxy-1,4-NQ, have been reported in the literature to possess redox activity. Although these molecules can act as electron acceptors, they do not exhibit electron transport functions. This leads to an unstable environment and the formation of free radicals, at the end, resulting in the disruption of signaling pathways and cellular proteins, DNA damage, and lipid degradation [33]. In this way, NADPH-cytochrome P450 reductase and flavoproteins in the respiratory chain can no longer function properly, and direct damage to the cell occurs [34,35]. From this point of view, it is suggested that the free radicals formed in *C. albicans*, although operating at very low concentrations of the compound (0.06 µgr/mL MIC_50_), can cause DNA damage, as well as membrane damage in *C. albicans*.

FT–IR spectra is a suitable method to understand the degeneration of cell wall lipids, proteins, and polysaccharides [29]. When comparing the FT–IR spectra of microorganisms treated with 5,8-dihydroxy-1,4-NQ with the untreated ones, the spectra overlapped, which indicated that no difference was observed. To obtain results at the FT–IR level, the 5,8-dihydroxy-1,4-NQ compound is expected to affect the covalent bonds of the cell wall and break various bonds on the cell wall surface [36]. Since this has not been demonstrated, the 5,8-dihydroxy-1,4-NQ compound may have formed temporary pores on the cell surface, resulting in an impairment of the permeability of the cell membrane and thus disrupting the ionic balance of the cell. Since covalent bonds do not need to be broken for antimicrobials to be effective, the chemical agent may have exerted its effect by forming pores in the cell membrane, just like the antimicrobial peptide nisin [37].

## 4. Materials and Methods

### 4.1. Screening of In Vitro Antimicrobial Effect

The antimicrobial assay of 5,8-dihydroxy-1,4-NQ, which was purchased from Apollo Scientific (USA), was performed against *C. albicans* as fungus; *E. coli*, *P. vulgaris*, *P. aeruginosa*, *K. pneumoniae*, and *S. enteritidis* as Gram-negative rods; *E. faecalis S. aureus*, and *S. epidermidis* as Gram-positive cocci; and *B. cereus* as Gram-positive rod control strains [38]. All fresh bacterial cultures were grown in Mueller Hinton Broth (MHB) and the fungus in Sabouraud Dextrose Broth (SDB). After 24 h of incubation, the fresh cultures were adjusted to 0.5 McFarland (about 10^8^ cfu/mL) using a turbidimeter (Becton Dickinson, USA). On the other hand, Mueller Hinton Agar (MHA) and Sabouraud Dextrose Agar (SDA) were prepared, and autoclaved sterile media were poured into Petri dishes, and wells were drilled on the agar surface. Fresh bacterial and fungal cultures were inoculated with a 100 μL sterile swap into the Petri dishes containing MHA and SDA, respectively. The concentration of 5,8-dihydroxy-1,4-NQ was adjusted to 1% (*w*/*v*) in DMSO, and 100 μL of the dissolved compound was added to the wells of the Petri dishes. Chloramphenicol (C^30^), a broad spectrum antibiotic, was used as a positive control (for *C. albicans*; nystatin 10,000 u/µL was used as a positive control). Samples with bacterial and fungal control strains on the media were incubated at 30 °C (for *C. albicans* 37 °C) for 24 h. The clear zones (no growth) around the wells were evaluated as antimicrobial activity against the control strains compared to the positive control zones.

### 4.2. Minimum Inhibition Concentration (MIC_50_) Assay

MIC_50_ values of the 5,8-dihydroxy-1,4-NQ compound were evaluated for those of the tested microorganisms that showed a distinct zone in the above experiment (susceptible strains). MIC_50_ assays were performed in 96-well microplates in MHB (SDB for *C. albicans*) in 12 serial dilutions in duplicate [39]. Serial dilutions from 310 μg/mL to 0.605 μg/mL in MHB in 96-well microplates were used for the MIC_50_ assay. On the other hand, the fresh bacterial cultures were adjusted to 0.5 McFarland using sterile MHB and a turbidimeter. Then, the fresh bacterial suspensions were added to the wells of the microplates in equal amounts (100 µL). All microplates were incubated at 30 °C (37 °C for fungi) for 24 h. After incubation, the absorbance value of the NQ dilutions was measured at 600 nm using a spectrophotometer (Shimadzu, UV 1800, Kyoto, Japan) with the susceptible strains along with the negative controls [blank medium (MHB or SDB without microorganisms)], the controls (fresh bacterial cultures without compound), and normalization controls (dilutions of the single compound) using a microplate reader (Fluostar Omega, BMG LABTECH, Cary, NC, USA). MIC_50_ assays were performed in 3 replicates for all susceptible strains.

### 4.3. Mechanism of Action Assays

#### 4.3.1. Preliminary Preparation of Experiments

The turbidity of fresh bacterial cultures (according to antimicrobial screening and MIC_50_ tests, only the susceptible ones) was adjusted to 0.5 McFarland. They were then inoculated at 2% (*v/v*) in Nutrient Broth (NB) and incubated overnight at 30 °C (for *C. albicans* 37 °C, in SDB). Then, cultures were centrifuged at 4000 rpm for 20 min at 4 °C. Supernatants were discarded, and cells were treated with their MIC_50_ concentration, mixed well, and incubated overnight at 30 °C overnight (for *C. albicans* 37 °C). Untreated cell suspensions in physiological saline solution (PSS) were used as negative controls. Then, treated and control cultures were centrifuged at 5000 rpm for 5 min. The pellets were used for crystal violet, TTC dehydrogenase activity, DNA damage, and SDS-PAGE assays. The supernatants of the pellets were used for DNA and protein leakage assays.

#### 4.3.2. Crystal Violet Assay

Permeability of cell membrane was demonstrated by the crystal violet uptake assay [40]. Pellets of selected microorganisms from the MIC assay were prepared as described previously and resuspended in phosphate buffer saline (PBS) containing 10 μg/mL crystal violet and incubated for 20 min at 30 °C. At the end of incubation, the cell suspensions were centrifuged at 5000 rpm for 15 min, and the absorbance of the supernatant at 590 nm absorbance was measured using a spectrophotometer. The crystal violet uptake results of all samples were calculated as a percentage using the following equation [41].
CVU = 100 − [(OD value of sample/OD value of control crystal violet) × 100](1)

Finally, the uptake of the crystal violet dye by the cells was evaluated by the percentage of empty crystal violet solution (the blank).

#### 4.3.3. TTC Dehydrogenase Activity

The metabolic activity of the 2, 3, 5-triphenyl tetrazolium chloride (TTC) dehydrogenase enzyme in cells was investigated by measuring the TTC dehydrogenase activity assay. Dehydrogenases, which are the load enzymes in the respiratory system, can easily penetrate the cell due to their low molecular weight. They oxidize an organic compound such as the insoluble red complex 2,3,5-triphenylformazan (TF) by taking up 2 hydrogens. TTC activity due to the presence and density of this molecule can be determined using a spectrophotometer at 485 nm absorbance [42]. Cells treated with 5,8-dihydroxy-1,4-NQ were prepared described above, and untreated cells were used as negative controls. Later, for measurement, 1 mL of 1.5 mol/L Tris–HCl buffer (pH 8.8) and 1 mL of TTC–glucose solution (prepared by mixing 4% TTC solution and 0.1 mol/L glucose solution in equal ratio) were added to the treated and control samples, which were resuspended in 3 mL of PSS. The mixture was incubated at 37 °C for 2 h. After incubation, methylbenzene and glacial acetic acid (GAA) (1:1, (3.75 mL)) was added to the samples. The mixture was kept at room temperature for 1 h to extract the organic phases. The upper phase of the samples was used to measure absorbance at 485 nm by using a spectrophotometer. Methylbenzene was used as a blank solution during the measurements. The following equation was used to calculate the RDA [43].
RDA (%) = (A_x_/A_0_) × 100 (2)

Herein, A_x_ indicates the absorbance of the treated samples and A_0_, the absorbance of the control sample.

#### 4.3.4. DNA Leakage

After treatment with the MIC_50_ value of 5,8-dihydroxy-1,4-NQ, the DNA leakage assays of the cells were evaluated. DNA leakage in the supernatant of the cells was measured using a spectrophotometer at 260 nm. The NQ derivative dissolved in PSS as a measure of the MIC_50_ concentration of each cell was used as a blank. When the 260 nm absorbance value was 1, the DNA concentration of the supernatants was considered to be 50 µg/mL [43].

#### 4.3.5. Protein Leakage

The leakage of protein from the cell supernatant indicates the presence of membrane damage to the microorganisms [43]. The leakage of proteins from damaged cells was studied when microorganisms were incubated with their MIC_50_ values 5,8-dihydroxy-1,4-NQ derivative. For this purpose, supernatants of cell suspensions treated with 5,8-dihydroxy-1,4-NQ were prepared as described above. Then, 500 μL of supernatant and 500 μL of the prepared Coomassie-Brilliant-Blue (CBB-250) solution were mixed and incubated in the dark at room temperature for 5 min. The absorbance of the samples was measured at 595 nm using a microplate reader. Standard BSA (bovine serum albumin) was used for the calibration curve [44].

#### 4.3.6. SDS PAGE Analysis

The changes in protein profile between control and NQ-treated cells were examined by SDS-PAGE analysis. Cells treated with 5,8-dihydroxy-1,4-NQ were prepared as described above, but at the beginning of the assay, cell cultures were grown overnight at 300 rpm and 30 °C by shaking to obtain sufficient protein extract for SDS analysis. Cells that were not treated with 5,8-dihydroxy-1,4-NQ were used as negative controls. Then the cells were centrifuged at 3500 rpm for 20 min; the supernatant was discarded again, and the pellets were dissolved in 1 mL sonication buffer and were sonicated in a Sonicator (Branson, Missouri, USA) for 20 s “on” and 15 s “off” at 20% amplitude for 5 min and lysed on ice. The lysed cells were centrifuged at 11,000 rpm for 30 min at 4 °C. After centrifugation, the supernatants were transferred to new centrifuge tubes; 5 µL sample of control cells was mixed with 250 µL Bradford reagent for protein extraction. Then, 10 µL from samples with the reagent was mixed with 5 µL of SDS loading buffer (with the addition of 3 µL dithiotheidol (DTT)) and incubated in a heat block for 5 min. The supernatant of the incubated samples was loaded onto a polyacrylamide gel and was run in separating gel at 90 V for about 1 h, followed by a run in stacking gel at 150V for about 1.5 h. Then, the gel was stained overnight with Coomassie Brilliant Blue R-250. Until decolorization, the gel was destained with water. Protein bands were visualized using the Gel Imager (Bio-Rad, Berkeley, California, USA). The Biolegend Protein Ladder was used as a marker.

#### 4.3.7. DNA Damage

SYBR Green I binds to ds DNA, which can be determined using a spectrophotometer with 525 nm absorbance [45]. Thus, when SYBR Green I dye has a combined effect on dsDNA, the absorbance value of the DNA of microorganisms decreases, and this is an indicator of DNA damage. Selected microorganisms were treated with 5,8-dihydroxy-1,4-NQ and prepared for experiments as described above. In addition, untreated microorganisms were used as negative controls. At this stage, cell cultures were inoculated at 4% (*v*/*v*) to obtain larger amounts of pellets/DNA of the selected microorganisms. Then, 20 µL of lysozyme (Sigma Aldrich, St. Louis, SI, USA) was added to the cell cultures. To increase the efficiency of the lysis buffer from PureLink^®^ Genomic DNA Mini Kits (Invitrogen, Waltham, MA, USA), 370 µL and 500 µL of bacteria and fungi cell suspensions, respectively, were added. They were then vortexed for 2 min and incubated at 37 °C for 1 h. For lysis control, the cell suspension treated with compound and negative controls was transferred to NA (nutrient agar) Petri dishes at a 100 µL volume and streaked with a Drigalski spatula. When no growth was observed in the inoculated Petri dishes, each treated sample, and the negative control were mixed with SYBR Green I (Sigma Aldrich, St. Louis, Missouri, USA) at a ratio of 1:1 and incubated at 37 °C for 15 min. Then, 200 µL samples from each mini centrifuge tube were transferred to the 96-well plate. Samples were measured in the microplate reader at 485 nm for excitation and 525 nm for emission. The absorbance values were compared with the absorbance value of the control well. If the absorbance value decreased compared to the control, it was considered DNA double-strand damage.

#### 4.3.8. FT–IR Analysis

For this assay, bacterial cells and *C. albicans* were prepared as described above. After the samples were centrifuged again at 4000 rpm for 30 min, the supernatants were discarded, and the pellets were dried by an evaporator (RC1010 Jouan, East Lyme, CT, USA). This analysis was applied by a Fourier transform–infrared spectrophotometer (FT–IR) (Perkin Elmer, Waltham, Massachusetts, USA) on the dried cell pellet. Graphics were plotted according to transmittance versus wavelengths (from 4000 to 600 cm^−1^). Untreated samples were used as negative controls.

The results of the FT–IR graphs of each sample and the control were overlaid on the computer using Origin-pro data analysis software. The change in peaks at different wavelengths of the treated samples compared to the control was evaluated [46].

#### 4.3.9. SEM Analysis

SEM analysis was used to show the changes in the cell morphology of the 3 control bacteria *(S. aureus*, *S. epidermidis*, and *C. albicans*) that had the highest membrane integrity according to the results of the crystal violet assay [47]. For SEM analysis, at least 10^7^ cell samples from each bacteria sample were treated with the selected bacterial MIC_50_ values (µg/mL) of the 5,8-dihydroxy-1,4-NQ. Untreated samples were used as negative controls. After overnight incubation, cells were washed 3 times with PSS and centrifuged at 4000 rpm for 30 min to remove the growth medium. After the last centrifugation, supernatants of the samples were discarded, and the pellets were mixed with 3 mL of the glutaraldehyde–PBS mixture (20 mL of PBS and 3 mL of glutaraldehyde). In the next step, they were incubated in the mixture at +4 °C overnight. Then they were centrifuged at 4500 rpm for 30 min and washed with 1 mL of PBS, and the pellets with 1 mL of PBS suspension were transferred to mini centrifuge tubes. Samples in the mini centrifuge tubes were re-centrifuged at 13,500 rpm for 15 min at +4 °C and washed 2 times to remove glutaraldehyde. The final cell pellets were re-suspended with PBS and visualized by SEM.

## 5. Conclusions

The present study demonstrated the antimicrobial activity of 5,8-dihydroxy-1,4-NQ on nine bacteria and one fungus. This commercial NQ was found to be effective against *S. aureus*, *S. epidermidis*, *S. enteritidis*, *B. cereus*, *P. vulgaris*, and *C. albicans*, with a very low MIC_50_. In addition, this study provided further information on the mechanism of the antimicrobial action of 5,8-dihydroxy-1,4-NQ, such as membrane integrity disruption associated with a decrease in respiratory activities, DNA damage, and leakage. Thus, this NQ could be considered a new potential candidate for antibacterial and antifungal agents that could inspire further discoveries in these microorganisms.

## Figures and Tables

**Figure 1 antibiotics-11-01537-f001:**
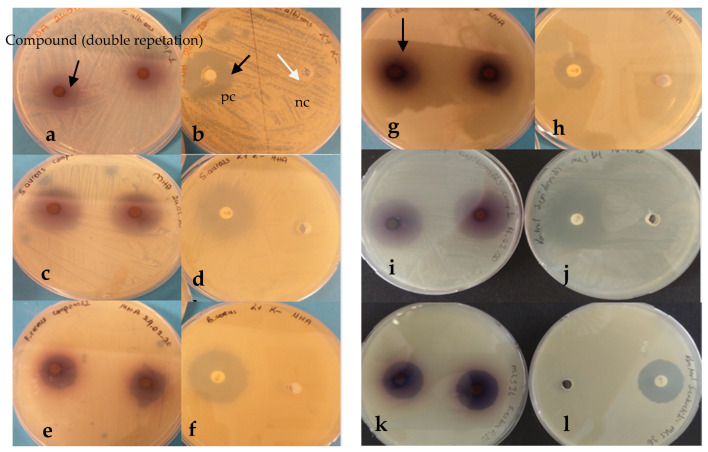
Antimicrobial screening test results showing antibacterial activity on the tested microorganisms; (**a**) *C. albicans* and (**b**) positive (pc) and negative (nc) controls; (**c**) *S. aureus* and (**d**) pc and nc; (**e**) *B. cereus*, (**f**) pc and nc; (**g**) *P. vulgaris* and (**h**) pc and nc; (**i**) *S. epidermidis*, (**j**) pc and nc; (**k**) *S. enteritidis*, and (**l**) pc and nc. Chloramphenicol (C^30^) disks were used as pc for all tested bacteria, whereas nystatin (10,000 u/µL) was utilized for yeast; nc was DMSO without compound. The black arrows indicate the antibacterial activity zone (inhibition of the growth of bacteria), and the white arrow indicates DMSO as nc on Petri dishes.

**Figure 2 antibiotics-11-01537-f002:**
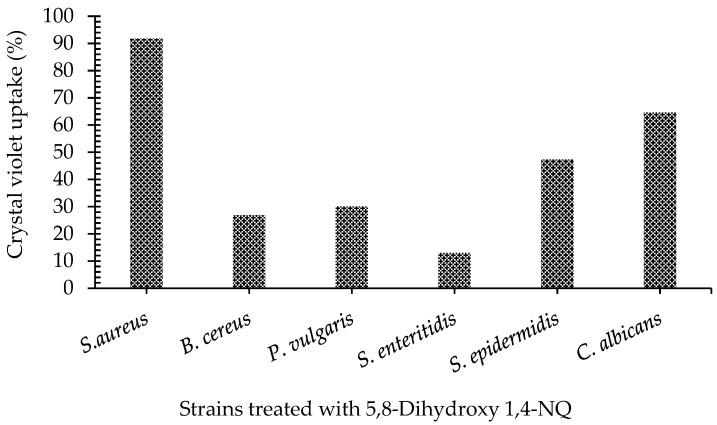
Ratio (%) of crystal violet uptake for all tested microorganisms.

**Figure 3 antibiotics-11-01537-f003:**
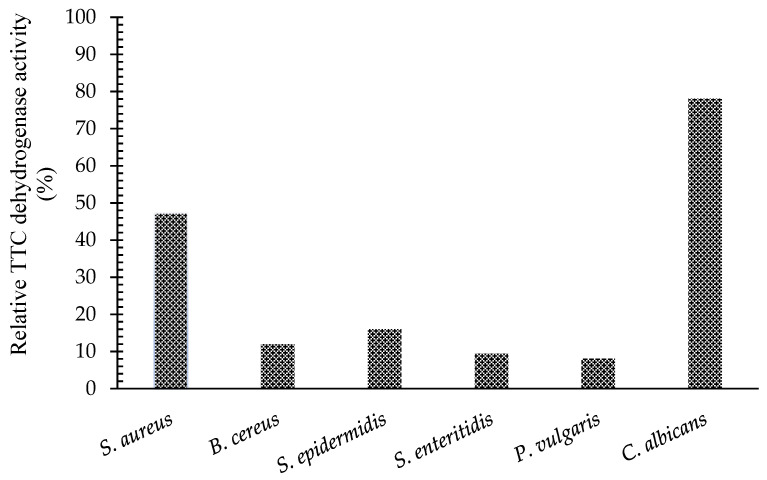
Relative TTC dehydrogenase activity (%).

**Figure 4 antibiotics-11-01537-f004:**
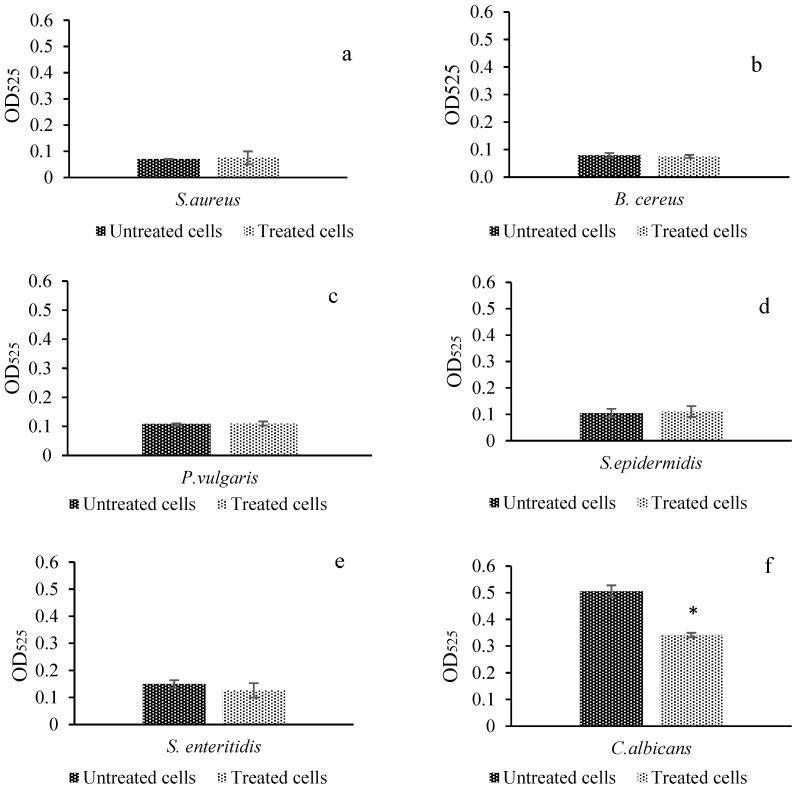
Results of DNA damage selected control microorganisms ((**a**). *S. aureus*; (**b**). *B. cereus*, (**c**). *P. vulgaris*; (**d**). *S. epidermidis*; (**e**). *S. enteritidis*; (**f**). *C. albicans* with SYBR Green I.) * indicates significant difference, *p* < 0.05.

**Figure 5 antibiotics-11-01537-f005:**
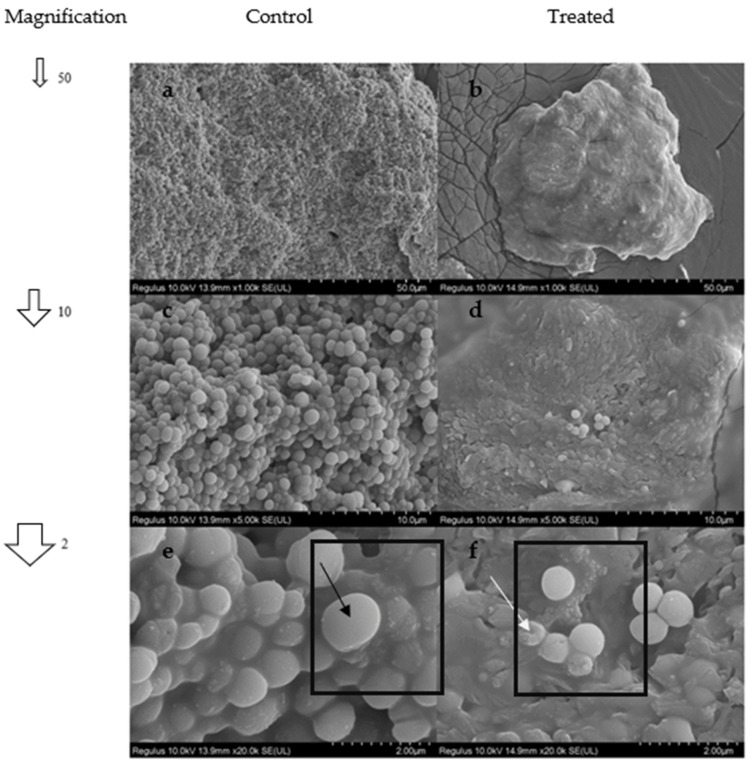
*S. aureus* SEM images from 50 to 2 µm ((**a**,**c**,**e**) (untreated negative control)/(**b**,**d**,**f**) (treated with compound), respectively). The black arrow in square (**e**) indicates a healthy membrane, while the white arrow in square (**f**) indicates the membrane damage of an individual cell.

**Figure 6 antibiotics-11-01537-f006:**
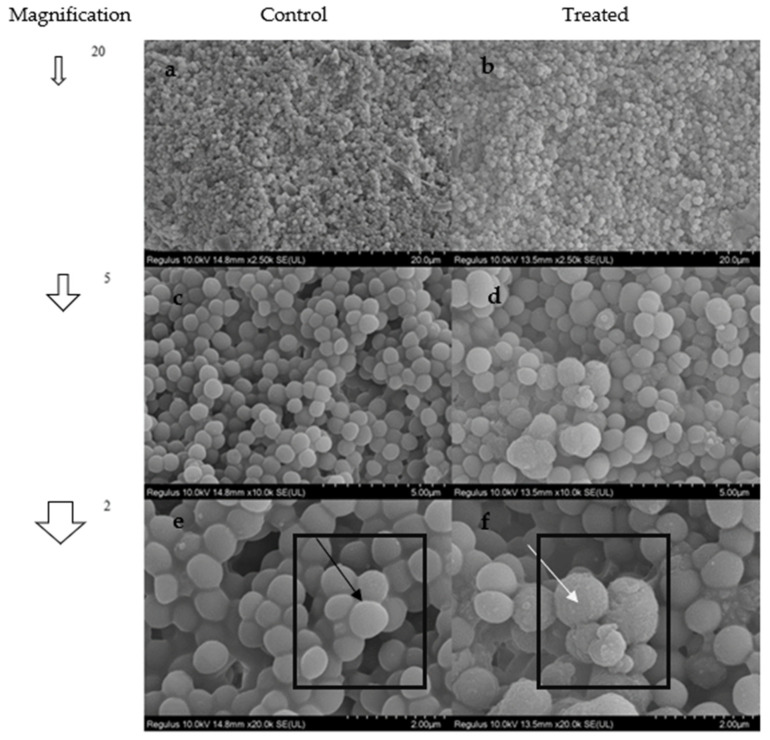
*S. epidermidis* SEM images from 50 to 2 µm ((**a**,**c**,**e**) (untreated)/(**b**,**d**,**f**) (treated)] respectively). The black arrow in square (**e**) indicates a healthy membrane, whereas the white arrow in square (**f**) indicates membrane damage at the cell surface.

**Figure 7 antibiotics-11-01537-f007:**
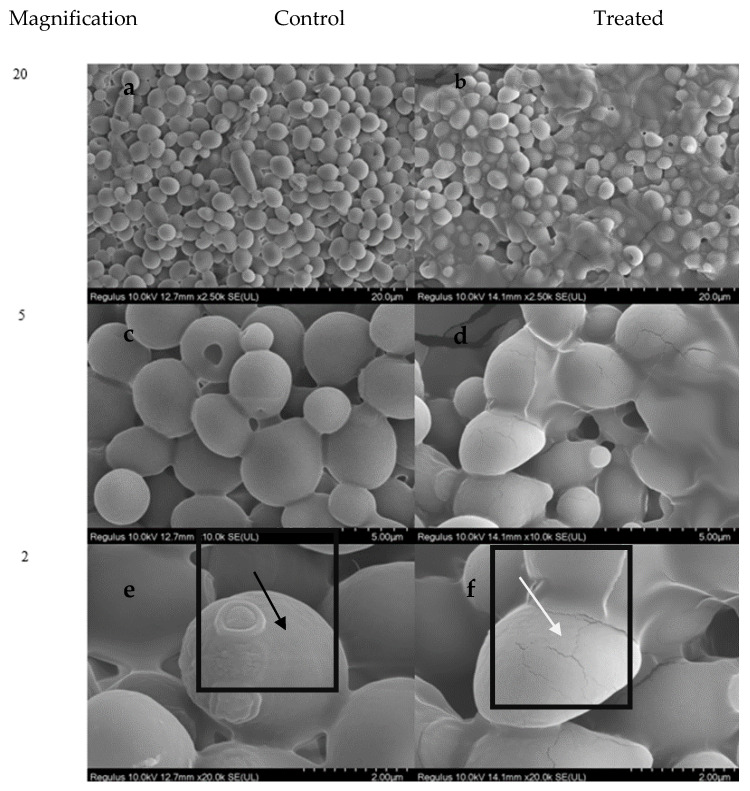
*C. albicans* SEM images from 50 to 2 µm ((**a**,**c**,**e**) (untreated negative control)/(**b**,**d**,**f**) (treated with compound) respectively). The black arrow in square (**e**) indicates a healthy membrane, while the white arrow in square (**f**) indicates damage to cell surface integrity.

**Table 1 antibiotics-11-01537-t001:** Antimicrobial assay results of tested microorganisms against 1% (*w*/*v*) 5,8-dihydroxy-1,4-NQs in DMSO.

Microorganism	Zone Diameter (mm) *	Negative Control DMSO	Positive Control Chloramphenicol (C^30^), Nystatin **
*S. aureus*	24	-	26 cm
*S. epidermidis*	30	-	26 cm
*E. faecalis*	11	-	19 cm
*B. cereus*	18	-	26 cm
*P. vulgaris*	19	-	18 cm
*E. coli*	8	-	25 cm
*K. pneumoniae*	-	-	22 cm
*P. aeruginosa*	-	-	15 cm
*S. enteritidis*	22	-	23 cm
*C. albicans*	18	-	25 cm

*** The largest zones are shown in bold. ** Positive control for fungi (*C. albicans*).

**Table 2 antibiotics-11-01537-t002:** MIC_50_ values of microorganisms.

Microorganism Name	MIC_50_ Value
*S. aureus*	1.2 μg/mL
*S. enteritidis*	6 μg/mL
*S. epidermidis*	2.4 μg/mL
*B. cereus*	14 μg/mL
*P. vulgaris*	10 μg/mL
*C. albicans*	<0.6 μg/mL

**Table 3 antibiotics-11-01537-t003:** DNA leakage of the sample supernatants.

Microorganisms	DNA Leakage (µg/mL)
*S. enteritidis*	4.375
*C. albicans*	3.25
*S. epidermidis*	2.875
*S. aureus*	ND *
*B. cereus*	ND
*P. vulgaris*	ND

* ND: not determined.

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
