# Peer review of "In Vitro Antimicrobial Effects and Inactivation Mechanisms of 5,8-Dihydroxy-1,4-Napthoquinone"

_antibiotics, 2022, doi:10.3390/antibiotics11111537_

Round 1
Reviewer 1 Report
-The introduction, abstract and conclusion should be improved to clarify the aim of the work
-English editing should be done
- The names of the microbes should be written italic in the conclusion
-The preparation procedures and full characterization of the Naphthoquinones (NQ) tested derivatives should be stated in the manuscript and also their chemical structure should be maintained.
Reviewer 2 Report
This paper entitled"In-vitro Antimicrobial Effects and Inactivation Mechanisms of 5,8 Dihydroxy 1,4-Naphthoquinone" is appeared to be a nice piece of work and will provide more information and reference for future study. However, I think this need minor revision to accept. As a commercial material, the author should introduce its use in more detail in the foreword, so as to better reflect the significance of this article. In addition, compared with providing data, it is more intuitive to respond to the bacteriostatic effect if relevant pictures of bacteriostasis experiments can be provided.
Reviewer 3 Report
The manuscript does not contain novel information; still, I think it will be good to have such studies published to expand the data in the field of antibiotics. However, I am deeply disappointed to see how the authors could not present their data in a proper scientific manner. The manuscript is full of typos and seems like it was not checked after preparing the first draft. This is not very common anymore to see such errors in the manuscript as several auto-checking modes are available in any digital writing program. On the technical side, in my opinion, some data and details are missing. Please see below some comments that can help to improve the article:
-
Table 1: Showing only zone diameter is not enough evidence. Please provide the images for better visual comparison.
-
Figures 4, 5, and 6 - These figures are not showing consistent differences between untreated and treated cells. The authors should explain precisely what these images reflect and how to read the differences there. Also, it will be effective to cite some papers here to show similar effects via SEM images. This will be beneficial for readers to indirectly validate these results.
-
The manuscript is written very poorly, and it looks like did not check for eliminating typos. Please see some examples below. I am not listing all of the errors here, so check the entire text carefully.
-
FT-IR Analizi ?
-
Figure 1: Ratio (%) of crystal violet uptake assay%91,83 ile (%) ?
-
Several different ways are used to cite figures “Figures 5, a and b”, “(5 f)”, “Figure 5e.”. This is a good example of noticing that manuscript was not checked carefully at all.
-
MIC50 or MIC50 Please be consistent with abbreviations.
-
and so on…
Author Response
Please see the attacment

Round 2
Reviewer 1 Report
The revision is sufficient, I recommended acceptance in the present form
Reviewer 3 Report
The authors did a great job in incorporating the previous comments. The current version of the manuscript reads well. I have no further major comments.